# Wheat Transcriptional Corepressor TaTPR1 Suppresses Susceptibility Genes *TaDND1/2* and Potentiates Post-Penetration Resistance against *Blumeria graminis forma specialis tritici*

**DOI:** 10.3390/ijms25031695

**Published:** 2024-01-30

**Authors:** Pengfei Zhi, Rongxin Gao, Wanzhen Chen, Cheng Chang

**Affiliations:** College of Life Sciences, Qingdao University, Qingdao 266071, China

**Keywords:** TOPLESS-related 1, *Defense no Death 1*, *Defense no Death 2*, bread wheat, powdery mildew

## Abstract

The obligate biotrophic fungal pathogen *Blumeria graminis forma specialis tritici* (*B.g. tritici*) is the causal agent of wheat powdery mildew disease. The TOPLESS-related 1 (TPR1) corepressor regulates plant immunity, but its role in regulating wheat resistance against powdery mildew remains to be disclosed. Herein, TaTPR1 was identified as a positive regulator of wheat post-penetration resistance against powdery mildew disease. The transient overexpression of *TaTPR1.1* or *TaTPR1.2* confers wheat post-penetration resistance powdery mildew, while the silencing of *TaTPR1.1* and *TaTPR1.2* results in an enhanced wheat susceptibility to *B.g. tritici.* Furthermore, *Defense no Death 1* (*TaDND1*) and *Defense no Death 2* (*TaDND2*) were identified as wheat susceptibility (*S*) genes facilitating a *B.g. tritici* infection. The overexpression of *TaDND1* and *TaDND2* leads to an enhanced wheat susceptibility to *B.g. tritici*, while the silencing of wheat *TaDND1* and *TaDND2* leads to a compromised susceptibility to powdery mildew. In addition, we demonstrated that the expression of *TaDND1* and *TaDND2* is negatively regulated by the wheat transcriptional corepressor TaTPR1. Collectively, these results implicate that TaTPR1 positively regulates wheat post-penetration resistance against powdery mildew probably via suppressing the *S* genes *TaDND1* and *TaDND2*.

## 1. Introduction

As the most widely cultivated cereal crop, allohexaploid bread wheat (*Triticum aestivum* L.) provides approximately 20% of the total calories in human food [1]. The world’s population is projected to reach 9.7 billion by 2050 and rise further to 11.2 billion in 2100, which drives the global demand for wheat grains [2]. However, the plant growth and global production of bread wheat are challenged by stressful environments, particularly invading pathogens and pests (P and Ps) [3]. Wheat powdery mildew disease caused by the pathogenic fungus *Blumeria graminis forma specialis tritici* (*B.g. tritici*) adversely affects the global wheat production [4,5]. Exploring the molecular mechanism underlying the wheat–*B. g. tritici* interaction and developing wheat varieties with an improved powdery mildew resistance are essential for controlling the powdery mildew epidemic and securing wheat production.

During the long-term coevolution, adapted pathogens and their host plants have acquired sophisticated strategies to facilitate their infection and defense, respectively. *Susceptibility* (*S*) genes from host plants are exploited by adapted pathogens to support the compatibility of the pathogens with plants probably via promoting pathogen (pre)penetration, suppressing plant immunity, and facilitating pathogen sustenance [6,7]. Upon the detection of invading pathogens, plants initiate two intertwined layers of induced defenses, pattern-triggered immunity (PTI) and effector-triggered immunity (ETI), to defend against pathogen infections [8,9,10,11,12,13,14,15,16,17,18]. During PTI and ETI, massive transcriptomic reprogramming is usually initiated, and this defense-related transcriptomic reprogramming is under the tight control of transcriptional regulators [19,20,21]. Identifying *S* genes and defense-related transcriptional regulators could deepen our understanding of the wheat–*B.g. tritici* interaction and assist wheat breeding for *B.g. tritici* resistance.

TOPLESS (TPL)/TOPLESS-related (TPR) transcriptional corepressors regulate plant development and environmental adaptation. In the model plant *Arabidopsis thaliana* (L.) Heynh, the transcriptional repressor AtAUX/IAA interacts with AtTPL to suppress the expression of auxin response factor (AtARF) target genes in the absence of auxin, whereas transcription factors BRI1-EMS-SUPPRESSOR 1 (AtBES1) and BRASSINAZOLE-RESISTANT 1 (AtBZR1) associate with the AtTPL-AtHDA19 complex to regulate the *Arabidopsis* brassinosteroid (BRs) signaling pathway [22,23]. There is increasing evidence showing that TPR1 plays a vital role in the regulation of plant immunity [24]. Indeed, knocking out *Arabidopsis AtTPR1* and its close homologs compromises the immunity mediated by the toll-like/interleukin-1 receptor (TIR)-NB-LRR R protein, a suppressor of npr1-1, constitutive 1 (AtSNC1), whereas the overexpression of *AtTPR1* constitutively activates AtSNC1-mediated immune responses [25,26]. Similarly, the silencing of *NbTPR1* in *Nicotiana benthamiana* compromised the flg22-triggered PTI defense response [27]. However, the potential function of wheat TPR1 homologs in the regulation of the wheat–*B.g. tritici* interaction is poorly understood.

*Arabidopsis S* genes *Defense no Death 1* (*AtDND1*) and *Defense no Death 2* (*AtDND2*) encode cyclic nucleotide-gated cation channels (CNGC; also known as AtCNGC2 and AtCNGC4, respectively). *Arabidopsis dnd1* and *dnd2* mutants exhibited a broad-spectrum disease resistance against a wide range of pathogens, including the bacterial pathogen *Pseudomonas syringae* pv. tomato and the oomycete pathogen *Hyaloperonospora parasitica* [28,29,30]. Similarly, the silencing of *StDND1* and *SlDND1*, *Arabidopsis AtDND1* orthologs, in potato and tomato crops, respectively, leads to an elevated resistance to late blight (*Phytophthora infestans*), powdery mildew (*Oidium neolycopersici* and *Golovinomyces orontii*), and grey mold (*Botrytis cinerea*) [31,32,33]. However, whether and how the wheat *DND1* and *DND2* homologs regulate the powdery mildew resistance remains unknown.

Herein, *TaTPR1.1* and *TaTPR1.2* are identified as positive regulators of wheat post-penetration resistance against powdery mildew disease. The transient overexpression of *TaTPR1.1* or *TaTPR1.2* confers wheat post-penetration resistance to powdery mildew, while the silencing of *TaTPR1.1* and *TaTPR1.2* results in an enhanced wheat susceptibility to *B.g. tritici.* Furthermore, *TaDND1* and *TaDND2* were identified as wheat *S* genes facilitating a *B.g. tritici* infection. The overexpression of *TaDND1* and *TaDND2* leads to an enhanced wheat susceptibility to *B.g. tritici*, while the silencing of wheat *TaDND1* and *TaDND2* leads to a compromised susceptibility to powdery mildew. In addition, we demonstrated that the expression of *TaDND1* and *TaDND2* is negatively regulated by the wheat transcriptional corepressor TaTPR1. This evidence strongly supports that TaTPR1 corepressors positively regulate wheat post-penetration resistance against powdery mildew by suppressing the expression of the *S* genes *TaDND1* and *TaDND2*. These findings could enhance our understanding of the genetic basis of wheat–*B.g. tritici* interactions and provide a new avenue for breeding wheat varieties with powdery mildew resistance.

## 2. Results

### 2.1. Homology-Based Identification of Wheat TaTPR1

In this study, a wheat homolog of *Arabidopsis* AtTPR1 was identified and characterized in the regulation of the wheat–*B.g. tritici* interaction. *TaTPR1.1* and *TaTPR1.2* were obtained from the reference genome of the hexaploid wheat by using the amino acid sequence of AtTPR1 (At1g80490) as a query. Three highly homologous sequences of *TaTPR1.1* genes separately located on chromosomes 4A, 4B, and 4D were obtained from the wheat genome sequence and designated as *TaTPR1.1-4A* (*TraesCS4A02G083300*), *TaTPR1.1-4B* (*TraesCS4B02G220900*), and *TaTPR1.1-4D* (*TraesCS4D02G221200*). Similarly, three highly homologous sequences of *TaTPR1.2* genes separately located on chromosomes 7A, 7B, and 7D were obtained from the wheat genome sequence and designated as *TaTPR1.2-7A* (*TraesCS7A02G296100*), *TaTPR1.2-7B* (*TraesCS7B02G189300*), and *TaTPR1.2-7D* (*TraesCS7D02G293500*).

As shown in Figure 1A, these predicted TaTPR1.1-4A, TaTPR1.1-4B, TaTPR1.1-4D, TaTPR1.2-7A, TaTPR1.2-7B, and TaTPR1.2-7D proteins shared over a 66% of their identities with *Arabidopsis* AtTPR1. The TaTPR1.1-4A, TaTPR1.1-4B, TaTPR1.1-4D, TaTPR1.2-7A, TaTPR1.2-7B, and TaTPR1.2-7D proteins all contain two conserved WD domains (WD40) (Figure 1B). The coding regions of these *TaTPR1.1* and *TaTPR1.2* genomic sequences all contain 25 exons and 24 introns (Figure 1C). Further phylogenetic analysis revealed that the TaTPR1.1-4A, TaTPR1.1-4B, TaTPR1.1-4D, TaTPR1.2-7A, TaTPR1.2-7B, and TaTPR1.2-7D proteins share over 70% of their identities with the AtTPR1, AtTPL, and rice OsTPR1 proteins (Figure 2). In contrast, AtTPR2 and AtTPR3 reside in the distinct ‘TPR2’ clade together with wheat TaTPR2-3A, TaTPR2-3B, TaTPR2-3D, and rice OsTPR2 (Figure 2).

### 2.2. TaTPR1 Potentiates Wheat Post-Penetration Resistance against Powdery Mildew

These *TaTPR1.1-4A*, *TaTPR1.1-4B*, *TaTPR1.1-4D*, *TaTPR1.2-7A*, *TaTPR1.2-7B*, or *TaTPR1.2-7D* genes were overexpressed in the leaf epidermal cells of the powdery mildew-susceptible wheat cultivar Yannong 999 using transient gene expression assays. After the inoculation of conidia from the virulent *B.g. tritici* isolate E09, the formation of *B.g. tritici* haustoria was statistically analyzed to evaluate the wheat post-penetration susceptibility to powdery mildew. As shown in Figure 3A, the *B.g. tritici* haustorium index (HI%) decreased from 58% for the empty vector (OE-EV) control to below 37% on wheat cells overexpressing *TaTPR1.1* or *TaTPR1.2* genes. These results suggested that the overexpression of *TaTPR1* could enhance the formation of *Bgt* haustoria and attenuate the wheat post-penetration susceptibility to the fungal pathogen *B.g. tritici*.

Thereafter, transiently induced gene silencing (TIGS) assays were performed to separately silence all endogenous *TaTPR1.1* or *TaTPR1.2* genes in the wheat epidermal cells. As shown in Figure 3B, the single silencing of *TaTPR1.1* or *TaTPR1.2* genes failed to cause a significant change in the HI%, compared to 38% for the empty vector (OE-EV) controls. In contrast, the simultaneous silencing of *TaTPR1.1* and *TaTPR1.2* could lead to a significant increase in the HI% to approximately 50%, suggesting that *TaTPR1.1* and *TaTPR1.2* might redundantly attenuate the formation of *Bgt* haustoria and contribute to the post-penetration resistance of wheat to *B.g. tritici* (Figure 3B).

To further verify the function of *TaTPR1* genes in the regulation of the wheat–*B.g. tritici* interaction, we employed barley stripe mosaic virus (BSMV)-induced gene silencing (BSMV-VIGS) to silence all endogenous *TaTPR1.1* or *TaTPR1.2* genes in the wheat leaves. A qRT-PCR assay demonstrated that the expression levels of *TaTPR1.1* or *TaTPR1.2* declined in the indicated VIGS plants (Figure 3C). After the inoculation of *B.g. tritici* conidia, the formation of microcolonies was statistically analyzed to evaluate the wheat post-penetration susceptibility to powdery mildew. As shown in Figure 3D, the microcolony index (MI%) increased to approximately 64% on *BSMV-TaTPR1.1as* + *BSMV-TaTPR1.2as* plants, compared with 55% for the *BSMV-γ* plants, 57% for the *BSMV-TaTPR1.1as* plants, and 54% for the *BSMV-TaTPR1.2as* plants (Figure 3D). These data confirm that *TaTPR1.1* and *TaTPR1.2* redundantly contribute to the post-penetration resistance of wheat to *B.g. tritici.*

### 2.3. Homology-Based Identification of TaDND1 and TaDND2 in Bread Wheat

Previous studies have revealed that the *Arabidopsis* transcriptional corepressor AtTPR1 targets the *S* genes *AtDND1* and *AtDND2* [1,2,3,12,13]. In this study, wheat homologs of Arabidopsis *AtDND1* and *AtDND2* were identified and characterized in the regulation of the wheat–*B.g. tritici* interaction. *TaDND1*, *TaDND2.1*, and *TaDND2.2* were obtained from the reference genome of the hexaploid wheat by using the amino acid sequences of *Arabidopsis* AtDND1 (At5g15410) and AtDND2 (AT5G54250) as queries. Three highly homologous sequences of *TaDND1* genes separately located on wheat chromosomes 5A, 5B, and 5D were obtained and designated as *TaDND1-5A* (*TraesCS5A02G395300*), *TaDND1-5B* (*TraesCS5B02G400100*), and *TaDND1-5D* (*TraesCS5D02G404600*). Three highly homologous sequences of *TaDND2.1* genes separately located on wheat chromosomes 3A, 3B, and 3D were obtained and designated as *TaDND2.1-3A* (*TraesCS3A02G316300*), *TaDND2.1-3B* (*TraesCS3B02G350500*), and *TaDND2.1-3D* (*TraesCS3D02G315000*). Similarly, three highly homologous sequences of *TaDND2.2* genes separately located on wheat chromosomes 1A, 1B, and 1D were obtained and designated as *TaDND2.2-1A* (*TraesCS1A02G321700*), *TaDND2.2-1B* (*TraesCS1B02G334100*), and *TaDND2.2-1D* (*TraesCS1D02G322000*).

As shown in Figure 4A, these predicted TaDND1-5A, TaDND1-5B, and TaDND1-5D proteins shared about 67% of their identities with *Arabidopsis* AtDND1. The TaDND1-5A, TaDND1-5B, and TaDND1-5D proteins all contain an ion transport (Ion_trans) domain (Figure 4B). The coding regions of these allelic *TaDND1* genomic sequences all contain five exons and four introns (Figure 4D). As shown in Figure 4E, these predicted TaDND2.1-3A, TaDND2.1-3B, TaDND2.1-3D, TaDND2.2-1A, TaDND2.2-1A, and TaDND2.2-1D proteins shared over 59% of their identities with *Arabidopsis* AtDND2. The TaDND2.1-3A, TaDND2.1-3B, TaDND2.1-3D, TaDND2.2-1A, TaDND2.2-1A, and TaDND2.2-1D proteins all contain an Ion_trans domain and a cyclic nucleotide-binding (cNMP binding) domain (Figure 4F). The coding regions of these allelic *TaDND2.1* genomic sequences all contain five exons and four introns, whereas the coding regions of allelic *TaDND2.2* genomic sequences all contained four exons and three introns (Figure 4F).

### 2.4. TaDND1 and TaDND2 Positively Contribute to the Wheat Susceptibility to B.g. tritici

To characterize the functions of *TaDND1* and *TaDND2* in the regulation of the wheat–*B.g. tritici* interaction, we first employed transient gene expression assays to overexpress *TaDND1-5A*, *TaDND1-5B*, *TaDND1-5D*, *TaDND2.1-3A*, *TaDND2.1-3B*, *TaDND2.1-3D*, *TaDND2.2-1A*, *TaDND2.2-1A*, or *TaDND2.2-1D* genes in the wheat leaf epidermal cell. As shown in Figure 5A, the HI% increased from 55% for the empty vector control (OE-EV) to over 67% on wheat cells overexpressing *TaDND1* or *TaDND2* genes. These results suggest that the overexpression of *TaDND1* or *TaDND2* significantly attenuates the formation of *Bgt* haustoria and potentiates the wheat post-penetration susceptibility to *B.g. tritici*.

Thereafter, we employed the TIGS assays to silence all endogenous *TaDND1* or *TaDND2* genes in the leaf epidermal cell of the *B.g. tritici*-susceptible wheat cultivar Yannong 999. As shown in Figure 5B, the silencing of *TaDND1* genes resulted in a notable HI% reduction to about 6%, compared to 36% for the empty vector controls. Although the silencing of the *TaDND2.1* or *TaDND2.2* genes failed to cause a significant change in the HI%, the simultaneous silencing of *TaDND2.1* and *TaDND2.2* could lead to a remarkable decrease in the HI% to approximately 9% (Figure 5B). These results suggest that the redundant *TaDND2.1* and *TaDND2.2* attenuate the formation of *Bgt* haustoria and contribute to the wheat post-penetration susceptibility to *B.g. tritici*.

In addition, we employed BSMV-VIGS to silence all endogenous *TaDND1*, *TaDND2.1*, or *TaDND2.2* genes in the leaves of the *B.g. tritici*-susceptible wheat cultivar Yannong 999 (Figure 5C). As shown in Figure 5D, the *B.g. tritici* MI% decreased to about 14% on the BSMV-TaDND1as plants, compared with 56% for the BSMV-γ plants. Although the silencing of the *TaDND2.1* or *TaDND2.2* genes failed to cause an obvious change in the MI%, the simultaneous silencing of *TaDND2.1* and *TaDND2.2* could lead to a significant decrease in the MI% to about 10% (Figure 5D). Collectively, these results support that *TaDND2.1* and *TaDND2.2* contribute to the wheat post-penetration susceptibility to the adapted fungal pathogen *B.g. tritici*.

### 2.5. TaTPR1 Is a Transcriptional Corepressor and Suppresses the Expression of TaDND1 and TaDND2

It has been demonstrated that *Arabidopsis* TPR1 functions as a transcriptional corepressor [13]. To quantify the transcriptional regulatory activities of TPR1 proteins, we performed the *Arabidopsis* leaf protoplast transfection assay. As shown in Figure 6A, the LucA ratio has decreased from 1 for the Gal4 DNA binding domain (DBD) control to less than 0.45 under the presence of DBD-TaTPR1.1-4A, DBD-TaTPR1.1-4B, DBD-TaTPR1.1-4D, DBD-TaTPR1.2-7A, DBD-TaTPR1.2-7B, or DBD-TaTPR1.2-7D, indicating that TaTPR1.1 and TaTPR1.2 proteins exhibit a transcriptional repressing activity.

To further confirm the regulation of *TaTPR1* on the expression of wheat *TaDND1* and *TaDND2* genes, we employed BSMV-VIGS to silence all endogenous *TaTPR1* genes, including *TaTPR1.1* and *TaTPR1.2* genes, in the leaves of the wheat cultivar Yannong 999. As shown in Figure 6B, the silencing of the *TaTPR1.1* and *TaTPR1.2* genes could lead to a significant increase in the expression levels of *TaDND1* and *TaDND2*, indicating that the transcriptional corepressor TaTPR1 negatively regulates the expression of *TaDND1* and *TaDND2*. Collectively, these results support the idea that the transcriptional corepressor TaTPR1 directly suppresses the expression of *TaDND1* and *TaDND2*.

## 3. Discussion

### 3.1. TaTPR1 Positively Regulates Wheat Powdery Mildew Immunity

In this study, six *AtTPR1* homologs (*TaTPR1.1-4A*, *TaTPR1.1-4B*, *TaTPR1.1-4D*, *TaTPR1.2-7A*, *TaTPR1.2-7B*, and *TaTPR1.2-7D*) were identified from bread wheat. The overexpression of *TaTPR1.1* or *TaTPR1.2* could confer wheat post-penetration resistance against *B.g. tritici*. Although the single silencing of *TaTPR1.1* or *TaTPR1.2* genes failed to pose a significant effect on haustorium development and microcolony formation of *B.g. tritici*, the simultaneous silencing of *TaTPR1.1* and *TaTPR1.2* led to a significantly compromised resistance against *B.g. tritici*, implicating that *TaTPR1.1* and *TaTPR1.2* redundantly contribute to the post-penetration resistance of wheat to *B.g. tritici*. Similarly, knocking out *AtTPR1* and its close homologs in *Arabidopsis* or the silencing of *NbTPR1* in *N. benthamiana* compromised the plant ETI and PTI [24,25,26]. It was recently demonstrated that the *Arabidopsis* TPR1 protein could reduce the detrimental effects associated with an activated transcriptional immunity [14]. It is therefore intriguing to examine the potential contribution of wheat TaTPR1 to mitigate the deleterious effects of induced immunity in future research. In addition, *Arabidopsis* transcription factors AtAUX/IAA, AtBES1, and AtBZR1 could interact with AtTPL, the homolog of TaTPR1, to regulate plant responses to auxin and BRs [22,23]. The potential effects of TaTPR1 overexpression on wheat plant development and yields need to be characterized in future research.

### 3.2. TaDND1 and TaDND2 Contribute to Wheat Powdery Mildew Susceptibility

Herein, three *AtDND1* homologs (*TaDND1-5A*, *TaDND1-5B*, and *TaDND1-5D*) and six *AtDND2* homologs (*TaDND2.1-3A*, *TaDND2.1-3B*, *TaDND2.1-3D*, *TaDND2.2-1A*, *TaDND2.2-1A*, and *TaDND2.2-1D*) were identified from bread wheat. Overexpressing *TaDND1* leads to an enhanced wheat susceptibility to powdery mildew, while the silencing of *TaDND1* confers wheat post-penetration resistance against powdery mildew, suggesting that TaDND1, resembling its homolog *AtDND1* in *Arabidopsis*, positively contribute to the wheat powdery mildew susceptibility. Similarly, the overexpression of *TaDND2.1* or *TaDND2.2* significantly potentiates a wheat powdery mildew susceptibility. Although the single knockdown of *TaDND2.1* or *TaDND2.2* failed to pose a significant effect on haustorium development and microcolony formation of *B.g. tritici*, the simultaneous silencing of *TaDND2.1* and *TaDND2.2* resulted in the significantly elevated resistance against *B.g. tritici*, implicating that *TaDND2.1* and *TaDND2.2* redundantly contribute to the wheat powdery mildew susceptibility. It was previously demonstrated that the knockout of *Arabidopsis AtDND1* and *AtDND2* or silencing the homologs of *AtDND1* in potatoes and tomatoes resulted in an elevated plant resistance against bacterial, fungal, and oomycete pathogens [28,29,30,31,32,33]. This study further confirmed the contribution of the wheat *S* genes *TaDND1* and *TaDND2* in facilitating the wheat–*B.g. tritici* interaction.

Previous studies have identified *S* genes governing multiple processes in the wheat–*B.g. tritici* interaction [7]. For instance, the S factors TaMLO, TaEDR1, TaPOD70, TaHDA6, TaHOS15, TaHDT701, and TaCAMTA2/3 negatively regulate wheat defense-related gene expression and suppress the wheat post-penetration resistance to *B.g. tritici* [34,35,36,37,38,39,40,41,42,43]. In *Arabidopsis*, mutations that attenuated SA biosynthesis or signaling (*sid2*, *npr1*, and *ndr1*) abolished the enhanced resistance of *dnd* mutants against the bacterial pathogen *P. syringae* and the oomycete pathogen *H. parasitica*, but not the fungal pathogen *B. cinerea* [44]. In contrast, the disruption of *Arabidopsis* ethylene signaling (*ein2*) partially attenuated the enhanced resistance to *B. cinerea* but not to *P. syringae* or *H. parasitica* [44]. Therefore, more experiments are needed to elucidate the molecular mechanisms underlying the resistance to *B.g. tritici* in *TaDND1-* or *TaDND2*-silenced wheat plants. In addition, the activation of plant defense usually results in a fitness cost. The yield penalty associated with *TaDND1* or *TaDND2* silencing needs to be characterized in future research.

There is increasing evidence demonstrating that the inactivation of *S* genes could reduce the compatibility of host plants with adapted pathogens and confer plant disease resistance [7,39,45,46,47,48,49,50]. For instance, the knockout of wheat *S* genes *TaMLO* and *TaEDR1* by genome editing system transcription activator-like effector nucleases (TALENs) enhances powdery mildew resistance, whereas the targeted knockout of *TaMLO* using clustered regularly interspaced short palindromic repeats (CRISPR)–Cas9 (CRISPR–associated 9) systems confers wheat powdery mildew resistance without a yield penalty [41,42,51]. Similarly, wheat *tamlo* mutant lines identified in the screen using targeting-induced local lesions in genomes (TILLING) techniques exhibited an enhanced resistance against *B.g. tritici* [39]. Therefore, it is intriguing to examine the potential of inactivating the *S* genes *TaDND1* and *TaDND2* via genome editing and TILLING techniques in the future when breeding for wheat powdery mildew resistance.

### 3.3. Transcriptional Corepressor TaTPR1 Suppresses Expression of TaDND1 and TaDND2

As demonstrated in the *Arabidopsis* protoplast transrepression assay, TaTPR1.1 and TaTPR1.2 proteins exhibit a transcriptional repressing activity. In addition, we showed that the silencing of *TaTPR1.1* and *TaTPR1.2* genes by BSMV-VIGS led to the potentiated expression of *TaDND1* and *TaDND2* in wheat leaves. These experiments indicate that the wheat transcriptional corepressor TaTPR1 suppresses the expression of *TaDND1* and *TaDND2*. Previous studies have demonstrated that *Arabidopsis* AtTPR1 is associated with the promoters of *AtDND1* and *AtDND2* genes and represses the expression of *AtDND1* and *AtDND2* [25]. Collectively, these studies strongly support that the suppression of *DND1* and *DND2* genes by TPR1 might be conserved among dicots and monocots. *Arabidopsis* AtTPR1 is demonstrated to associate with histone deacetylase (HDAC) 19 [25]. Although whether the wheat TaTPR1 protein interacts with HDACs remains unknown, there is increasing evidence demonstrating that wheat HDACs are involved in the regulation of wheat powdery mildew resistance [52,53]. For instance, the RPD3 (reduced potassium dependency protein 3)-type HDAC TaHDA6 and the HD2 (histone deacetylase 2)-type HDAC TaHDT701 negatively regulate wheat defense to *B.g. tritici* by mediating histone deacetylation at the promoter regions of defense-related genes [52,53]. Identifying wheat HDACs associated with TaTPR1 might shed light on the molecular mechanism underlying TaTPR1’s function in the wheat–*B.g. tritici* interaction in future research.

## 4. Materials and Methods

### 4.1. Plant and Pathogen Materials

One wheat genotype, *B.g. tritici*-susceptible wheat cultivar Yannong 999, was employed in this study. Wheat seeds were surface sterilized and kept in pots containing soil in the greenhouse under a 16 h/8 h, 20 °C/18 °C day/night cycle with a 70% relative humidity. *A. thaliana* ecotype Columbia (Col-0) was used in this study. *A. thaliana* seeds were surface sterilized and kept in pots containing soil in a growth chamber under a 16 h/8 h light period at 23 °C with a 70% relative humidity. One *B.g. tritici* genotype, virulent *B.g. tritici* isolate E09, was used in this study. The *B.g. tritici* was maintained on the leaves of Yannong 999 wheat plants and kept at a 70% relative humidity and a 20 °C day/18 °C night cycle. The *B.g. tritici* inoculation and maintenance were performed as described previously [34].

### 4.2. Quantitative Reverse-Transcription PCR (qRT-PCR)

Total RNA was extracted using the TRIzol Reagent (Invitrogen, Carlsbad, CA, USA). The RNA quality was examined according to previous studies [54,55]. Two μg of total RNA was used to generate the cDNA template under the TransScript one-step gDNA removal and cDNA synthesis supermix according to the manufacturer’s instructions. The real-time PCR assay was performed using a qPCR master mix (Invitrogen). The *TaGADPH* gene was employed as the internal control, and the expressions of *TaTPR1.1*, *TaTPR1.2*, *TaDND1*, *TaDND2.1*, and *TaDND2.2* were analyzed using the primers 5′-ATCATTAAAACTAGGTGAT-3′/5′-GGCCTCATCAGGACTATTG-3′, 5′-GCATTTTCTCAATCAATG A-3′/5′-GCAGTGCATCTCTTGGGTA-3′, 5′-ATGCCTCCATCGCTCTCCT-3′/5′-GGCTGCGTGCACGCGTAAC-3′, 5′-TCCTCGCCTTCTTCCTCGT-3′/5′-CTTGGACCTCGGCAGCCGA-3′, and 5′-CGGCCACGGCGGTTGC GCG-3′/5′-CGGATCATCGCCGGCGCCG-3′, respectively. For the qRT-PCR, three independent biological replicates were statistically analyzed (*t*-test; * *p* < 0.05, ** *p* < 0.01) for each treatment. The qRT-PCR analysis experiments were repeated three times with similar results.

### 4.3. BSMV-Mediated Gene Silencing and Microcolony Index Analysis

For the BSMV-mediated gene silencing assay, antisense fragments of *TaTPR1.1*, *TaTPR1.2*, *TaDND1*, *TaDND2.1*, and *TaDND2.2* were cloned into the pCa-γbLIC vector using the primers 5′-AAGGAAGTTTAGCGGGTAGCTATGGCTCTGC-3′/5′-AACCACCACCACCGTTGGACCCTTTCAACCTGCAC-3′, 5′-AAGGAAGTTTAGTGCGAACAACTTGTTTGG-3′/5′-AACCACCACCACCGTTGGTTGGATGACAAATCCCA-3′, 5′-AAGGAAGTTTACATAAGCAAAGGCGCCATTG-3′/5′-AACCACCACCACGTTCATTGCCTCTCATATTGCA-3′, 5′-AAGGAAGTTTAGCCCGATCGCCGCCAGCCG-3′/5′-AACCACCACCACCGTGACCGACCTCTCGGCGTCG-3′, and 5′-AAGGAAGTTTAGCCCCAGCCCCAGCTGCTG-3′/5′-AACCACCACCACCGTGCTCTCCACGCGCTCGTCG-3′. The BSMV-mediated gene silencing assay and microcolony index (MI) analysis were performed as described previously [56]. At least 2000 wheat-*Bgt* interaction sites were counted in one experiment for each treatment, and three independent biological replicates were statistically analyzed (*t*-test; * *p* < 0.05, ** *p* < 0.01) for each treatment. The MI analysis experiments were repeated three times with similar results.

### 4.4. Single-Cell Transient Gene Silencing/Overexpression Assays and Haustorium Index Analysis

For the single-cell transient gene silencing assay, antisense fragments of *TaTPR1.1*, *TaTPR1.2*, *TaDND1*, *TaDND2.1*, and *TaDND2.2* were cloned into the pIPKb007 vector using the primers 5′-GGGGACAAGTTTGTACAAAAAAGCAGGCTTCGCGGGTAGCTATGGCTCTGC-3′/5′-GGGGACCACTTTGTACAAGAAAGCTGGGTCTGACCCTTTCAACCTGCAC-3′, 5′-GGGGACAAGTTTGTACAAAAAAGCAGGCTTCGTGCGAACAACTTGTTTGG-3′/5′-GGGGACCACTTTGTACAAGAAAGCTGGGTCTGGTTGGATGACAAATCCCA-3′, 5′-GGGGACAAGTTTGTACAAAAAAGCAGGCTTCCATAAGCAAAGGCGCCATTG-3′/5′-GGGGACCACTTTGTACAAAAAGCTGGGTCTCATTGCCTCTCATATTGCA-3′, 5′-GGGGACAAGTTTGTACAAAAAA GCAGGCTTCGCCCGATCGCCGCCAGCCG-3′/5′-GGGGACCACTTTGTACAAGAAAGCTGGGTCGACCGACCTCTCGGCGTCG-3′, and 5′-GGGGACAAGTTTGTACAAAAAAGCAGGCTTCGCCCCAGCCCCAGCTGCTG-3′/5′-GGGGACCACTTTGTACAAGAAAGCTGGGTCGCTCTCCACGCGCTCGTCG-3′, respectively. For the single-cell transient gene overexpression assay, coding regions of *TaTPR1.1-4A*, *TaTPR1.1-4B*, *TaTPR1.1-4D*, *TaTPR1.2-7A*, *TaTPR1.2-7B*, *TaTPR1.2-7D*, *TaDND1-5A*, *TaDND1-5B*, *TaDND1-5D*, *TaDND2.1-3A*, *TaDND2.1-3B*, *TaDND2.1-3D*, *TaDND2.2-1A*, *TaDND2.2-1A*, and *TaDND2.2-1D* were cloned into the pIPKb001 vector using the primers 5′-GGGGACAAGTTTGTACAAAAAAGCAGGCTTCATGTCTTCTCTCAGCCGGGA-3′/5′-GGGGACCACTTTGTACAAGAAAGCTGGGTCTTATCTTTCTGGTTGATCAGA-3′ (for amplifying coding regions of *TaTPR1.1-4A*, *TaTPR1.1-4B*, and *TaTPR1.1-4D*), 5′-GGGGACAAGTTTGTACAAAAAAGCAGGCTTCATGTCGTCGCTCAGCAGGGA-3′/5′-GGGGACCACTTTGTACAAGAAAGCTGGGTCTCATCTCGTTGGCTGATCAGA-3′ (for amplifying coding regions of *TaTPR1.2-7A*, *TaTPR1.2-7B*, and *TaTPR1.2-7D*), 5′-GGGGACAAGTTTGTACAAAAAAGCAGGCTTCATGCCTCCATCGCTCTCCTC-3′/5′-GGGGACCACTTTGTACAAGAAAGCTGGGTCCTACTCGAGGTGGTCGTGCG-3′ (for amplifying coding regions of *TaDND1-5A*, *TaDND1-5B*, and *TaDND1-5D*), 5′-GGGGACAAGTTTGTACAAAAAAGCAGGCTTCATGCCGACCGACCTCTCGGCGT-3′/5′-GGGGACCACTTTGTACAAGAAAGCTGGGTCTCAGAGCAGGAGGTCGTCCTG-3′ (for amplifying coding regions of *TaDND2.1-3A*, *TaDND2.1-3B*, and *TaDND2.1-3D*), and 5′-GGGGACAAGTTTGTACAAAAAAGCAGGCTTCATGTCCGGCGAGCTCTCCAC-3′/5′-GGGGACCACTTTGTACAAGAAAGCTGGGTCCTAGAAGGAGAAGTCGTCGTC-3′ (for amplifying coding regions of *TaDND2. 2-1A*, *TaDND2.2-1A*, and *TaDND2.2-1D*), respectively. The single-cell transient gene silencing/overexpression assays and haustorium index (HI) analysis were performed as described [34]. At least 100 cells were analyzed in one experiment, and three independent biological replicates were statistically analyzed (*t*-test; * *p* < 0.05, ** *p* < 0.01) for each treatment. The HI analysis experiments were repeated three times with similar results.

## 5. Conclusions

In this study, we characterized the function of wheat TaTPR1 in the regulation of the wheat–*B.g. tritici* interaction and demonstrated that *TaTPR1.1* and *TaTPR1.2* positively contribute to the wheat post-penetration resistance against *B.g. tritici.* The overexpression of *TaTPR1.1* or *TaTPR1.2* confers wheat post-penetration resistance against *B.g. tritici*, while the silencing of *TaTPR1.1* and *TaTPR1.2* results in a compromised wheat resistance against *B.g. tritici.* Furthermore, we found that *TaDND1* and *TaDND2* function as wheat *S* genes contributing to the wheat powdery mildew susceptibility. The knockdown of *TaDND1* or *TaDND2* expression using transient- or virus-induced gene-silencing attenuates the post-penetration susceptibility to *B.g. tritici*. In addition, we demonstrated that the expression of *TaDND1* and *TaDND2* is negatively regulated by the wheat transcriptional corepressor TaTPR1. These results collectively suggest that TaTPR1 positively regulates the wheat post-penetration resistance against *B.g. tritici* probably via suppressing the *S* genes *TaDND1* and *TaDND2*. These findings could enhance our understanding of the genetic basis of wheat–*B.g. tritici* interactions and promote breeding programs for future wheat varieties with an enhanced powdery mildew resistance.

## Figures and Tables

**Figure 1 ijms-25-01695-f001:**
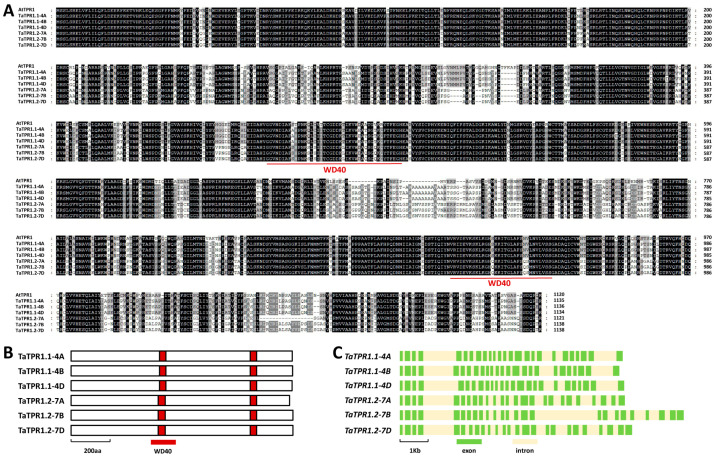
Identification of wheat TaTPR1 based on homology with *Arabidopsis* AtTPR1. (**A**) Protein sequence alignments of wheat TaTPR1.1, TaPRR1.2, and *Arabidopsis* AtTPR1. Identical residues among 7 protein sequences are shaded in dark, while residues conserved in at least 4 of the 7 proteins are shaded in gray. (**B**) Domain structures of wheat TaTPR1.1 and TaTPR1.2 proteins. (**C**) Gene architectures of wheat *TaTPR1.1* and *TaTPR1.2* genes.

**Figure 2 ijms-25-01695-f002:**
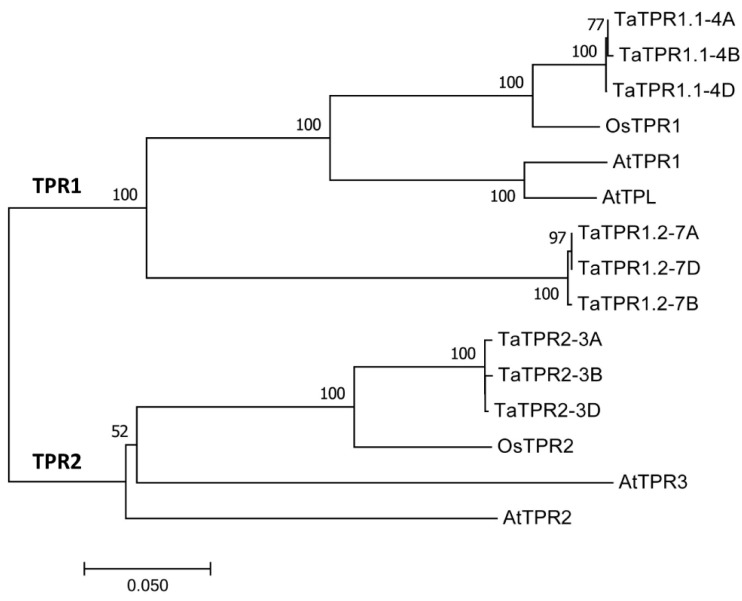
Phylogenetic relationships of the TPR1 and TPR2 homologs in *Arabidopsis*, rice, and bread wheat. The phylogenetic tree was constructed using the neighbor-joining method with 1000 bootstraps.

**Figure 3 ijms-25-01695-f003:**
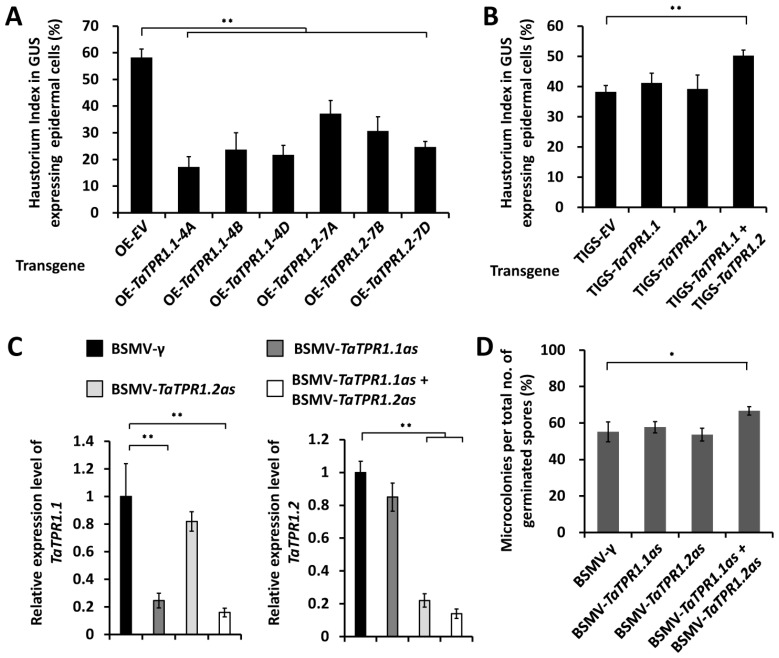
Functional analyses of *TaTPR1* genes in wheat–*Bgt* interaction. (**A**) Haustorial formation analysis in wheat epidermal cells transiently overexpressing *TaTPR1.1* (*OE-TaTPR1.1*) and *TaTPR1.2* (*OE-TaTPR1.2*). Haustorium index (HI%) on wheat epidermal cells bombarded with empty vector (*OE-EV*) was statistically analyzed as a control. More than 50 wheat cells were analyzed for each experiment. (**B**) Haustorial formation analysis in wheat epidermal cells transiently silencing *TaTPR1.1* (*TIGS-TaTPR1.1*) and *TaTPR1.2* (*TIGS-TaTPR1.2*) or cosilencing *TaTPR1.1* and *TaCAMTA3* (*TIGS-TaCAMTA2* + *TIGS-TaCAMTA3*). (**C**) qRT-PCR analysis of *TaTPR1.1* and *TaTPR1.2* expression in the wheat leaves infected with indicated BSMV vectors. BSMV-*γ* empty vector was employed as the negative control. (**D**) *Bgt* microcolony index analysis on wheat leaves silencing *TaTPR1.1* (*BSMV-TaTPR1.1as*) and *TaTPR1.2* (*BSMV-TaTPR1.2as*) or cosilencing *TaTPR1.1* and *TaTPR1.2* (*BSMV-TaTPR1.1as* + *BSMV-TaTPR1.2as*). For (**A**–**D**), three independent biological replicates were statistically analyzed for each treatment (*t*-test; * *p* < 0.05, ** *p* < 0.01).

**Figure 4 ijms-25-01695-f004:**
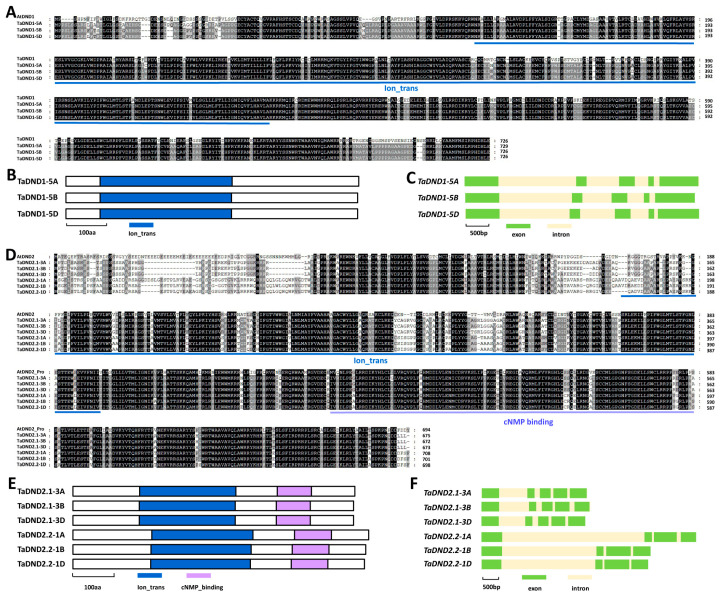
Identification of wheat TaDND1 and TaDND2 based on homology with *Arabidopsis* AtDND1 and AtDND2. (**A**) Sequence alignments of wheat TaDND1 and *Arabidopsis* AtDND1 proteins. Residues conserved in at least 2 of the 4 proteins are shaded in gray, while identical residues among 4 protein sequences are shaded in dark. (**B**) Domain structures of wheat TaDND1 proteins. (**C**) Gene architectures of wheat *TaDND1* genes. (**D**) Sequence alignments of wheat TaDND2.1, TaDND2.2, and *Arabidopsis* AtDND2 proteins. Residues conserved in at least 3 of the 6 proteins are shaded in gray, while identical residues among 6 protein sequences are shaded in dark. (**E**) Domain structures of wheat TaDND2.1 and TaDND2.2 proteins. (**F**) Gene architectures of wheat *TaDND2.1* and *TaDND2.2* genes.

**Figure 5 ijms-25-01695-f005:**
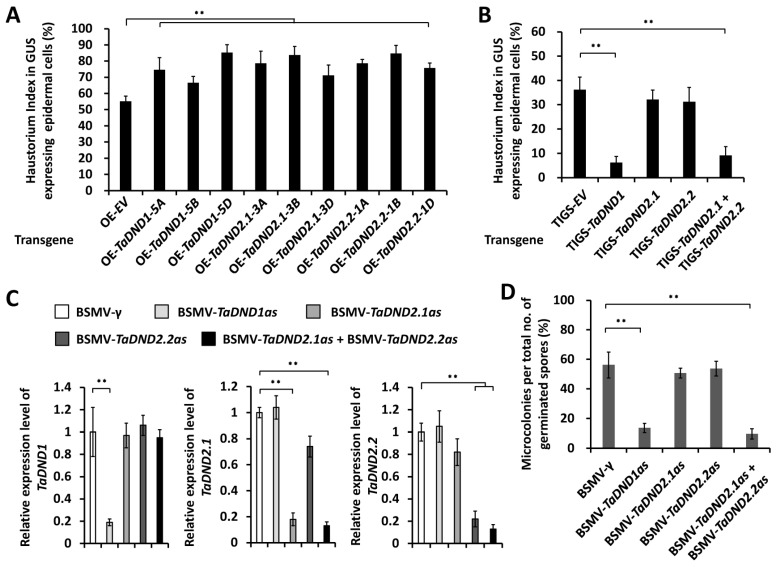
Functional analyses of *TaDND1* and *TaDND2* genes in wheat–*Bgt* interaction. (**A**) Haustorial formation analysis in wheat epidermal cells transiently overexpressing *TaDND1* (*OE-TaDND1*), *TaDND2.1* (*OE-TaDND2.1*), and *TaDND2.2* (*OE-TaDND2.2*). (**B**) Haustorium index analysis in wheat epidermal cells transiently silencing *TaDND1* (*TIGS-TaDND1*), *TaDND2.1* (*TIGS-TaDND2.1*), and *TaDND2.2* (*TIGS-TaDND2.2*) or cosilencing *TaDND2.1* and *TaDND2.2* (*TIGS-TaDND2.1*+ *TIGS-TaDND2.2*). (**C**) qRT-PCR analysis of *TaDND1*, *TaDND2.1*, and *TaDND2.2* expressions in the wheat leaves infected with indicated BSMV vectors. (**D**) *Bgt* microcolony index analysis on wheat leaves silencing *TaDND1* (*BSMV-TaDND1as*), *TaDND2.1* (*BSMV-TaDND2.1as*), and *TaDND2.2* (*BSMV-TaDND2.2as*) or cosilencing *TaDND2.1* and *TaDND2.2* (*BSMV-TaDND2.1as* + *BSMV-TaDND2.2as*). For (**A**–**D**), three independent biological replicates were statistically analyzed for each treatment (*t*-test; ** *p* < 0.01).

**Figure 6 ijms-25-01695-f006:**
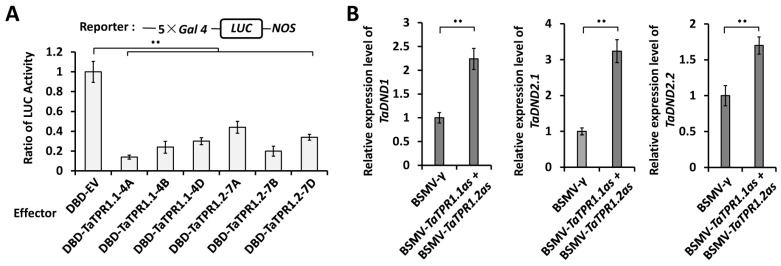
Analysis of the transcriptional suppression of *TaDND1* and *TaDND2* genes by TaTPR1. (**A**) Transcriptional repression activity analysis of TaTPR1.1 and TaTPR1.2 in *Arabidopsis* protoplast cells. (**B**) qRT-PCR analysis of *TaDND1* and *TaDND2* expression levels in *TaTPR1*-silenced wheat leaves. For (**A**) and (**B**), three independent biological replicates were statistically analyzed for each treatment (*t*-test; ** *p* < 0.01).

## Data Availability

Data presented here are available on request from correspondence.

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
