# Peer review of "Wheat Transcriptional Corepressor TaTPR1 Suppresses Susceptibility Genes TaDND1/2 and Potentiates Post-Penetration Resistance against Blumeria graminis forma specialis tritici"

_ijms, 2024, doi:10.3390/ijms25031695_

Round 1
Reviewer 1 Report (Previous Reviewer 2)
Comments and Suggestions for Authors
The article “Wheat Transcriptional CorepressorTaTPR1 Suppresses 2 Susceptibility Genes TaDND1/2 and Potentiates Post Pene- 3 tration Resistance against Blumeria graminis forma specialis tritici” (ijms- 2819635) studies the role of a transcriptional corepressor on wheat resistance to powdery mildew. The research appears to be well-conducted, but there are some minor issues to be addressed. Overall, the writing is ok, but there are some English writing mistakes, for example in line 24 of the abstract (it should be “suppressing” instead of "supperssing").
Lines 415-418: this sentence should be in the Materials and Methods section.
Pages 7-9: primers could be listed as a supplementary Table instead of in the Materials and Methods sections.
Please see Instructions for Authors regarding the placement of the figures in the manuscript. Figures should not be placed at the end of the text, they should be placed in the main text of the manuscript, soon after being mentioned.
Comments on the Quality of English LanguageMinor English writing mistakes.
Author Response
Reviewer 1 Comments for the Author...
The article “Wheat Transcriptional Corepressor TaTPR1 Suppresses Susceptibility Genes TaDND1/2 and Potentiates Post Penetration Resistance against Blumeria graminis forma specialis tritici” (ijms- 2819635) studies the role of a transcriptional corepressor on wheat resistance to powdery mildew. The research appears to be well-conducted, but there are some minor issues to be addressed.
- Overall, the writing is ok, but there are some English writing mistakes, for example in line 24 of the abstract (it should be “suppressing” instead of "supperssing").
- Response: Thank you very much for these encouraging comments. We have made extensive have made extensive revision of English writing according to referees’ comments. Hopefully, this version could meet the standard for publication.
- Lines 415-418: this sentence should be in the Materials and Methods section.
- Response: We fully agree with the Reviewer.. This sentence has been placed into the Materials and Methods section in the revision.
- Pages 7-9: primers could be listed as a supplementary Table instead of in the Materials and Methods sections.
- Response: Many thanks. Primers used in the study have been presented in a more readable way in the revised manuscript.
- Please see Instructions for Authors regarding the placement of the figures in the manuscript. Figures should not be placed at the end of the text, they should be placed in the main text of the manuscript, soon after being mentioned.
- Response: Thank you very much. Figures are currently placed at the end of the text for the convenience of review. After acceptance, this manuscript would underdo typesetting and these figures would be placed in the main text of the manuscript then.
Reviewer 2 Report (New Reviewer)
Comments and Suggestions for Authors
1. Line 157 - 159: “we are interested to …, we first searched…” could be changed to “wheat homolog of Arabidopsis AtTPR1 was identified and characterized…” A first-person narration in a research paper may not be good.
2. Line 164: The wheat CSv3 has been published. I suggest using the CSv3 reference.
3. Line 178: “… are closely related to the AtTPR1, AtTPL and rice OsTPR1”. I recommend mentioning that how close they are in the text. Not just cite the fig.
4. Line 264: “As shown in Figure 3D, the microcolony index (MI%) increased …” Here the author used MI to evaluate the resistance/susceptibility in VIGS plants. Why not continuously use the HI index as same as before? Does HI index in the VIGS plants show significant difference? How is the difference?
5. Line 411-413: “the expression of DBD-TaTPR1.1-4A, DBD-TaTPR1.1-4B, DBD-TaTPR1.1-4D, DBD-TaTPR1.2-7A, DBD-TaTPR1.2-7B, or DBD-TaTPR1.2-7D…” Using “expression of …”, we always mean the expression of genes and should use italics here. Or just delete “the expression of” to refer the protein.
6. Line 696: Didn’t see any instruction about haustorium index analysis. Also, I suggest giving a brief introduction about the MI index in “4.3. BSMV-mediated gene silencing and microcolony index analysis”.
7. Are there any figures of wheat leaves between OE/VIGS and the control? A picture of leaves may give reader a directly impression about how the resistance or susceptibility is.
8. TaTPR related to auxin. Overexpression of TaTPR could enhance resistance. Does the overexpression make any yield difference?
9. The susceptible genes in the hosts often play positive or critical roles in plant development, and silencing or deletion of those S genes may have negative effects on plant development or yield. Is there any fitness cost of TaDND?
Author Response
Reviewer 2 Comments for the Author...
- Line 157 - 159: “we are interested to …, we first searched…” could be changed to “wheat homolog of Arabidopsis AtTPR1 was identified and characterized…” A first-person narration in a research paper may not be good.
- Response: We thank the Reviewer for this excellent suggestion. We have replaced “we are interested to …, we first searched…” with “wheat homolog of Arabidopsis AtTPR1 was identified and characterized…” in the revised manuscript.
- Line 164: The wheat CSv3 has been published. I suggest using the CSv3 reference.
- Response: Many thanks. The wheat CSv3 reference has been cited in the new version
- Line 178: “… are closely related to the AtTPR1, AtTPL and rice OsTPR1”. I recommend mentioning that how close they are in the text. Not just cite the fig.
- Response: Many thanks! Sequence identity among TaTPL1, AtTPR1, AtTPL and rice OsTPR1 has been mentioned in the revised manuscript.
- Line 264: “As shown in Figure 3D, the microcolony index (MI%) increased …” Here the author used MI to evaluate the resistance/susceptibility in VIGS plants. Why not continuously use the HI index as same as before? Does HI index in the VIGS plants show significant difference? How is the difference?
- Response: Thank you for this suggestion. The microcolony index (MI%) describes the fungal microcolony formation, whereas the haustrial index (HI%) describes the fungal haustorial formation formation. Therefore, MI index and HI index were analyzed to show two different post-penetration events. Herein, HI index was measured in the single cell transient gene expression/silencing assays instead of VIGS, which is a routine experiment in the analysis of wheat powdery mildew resistance.
- Line 411-413: “the expression of DBD-TaTPR1.1-4A, DBD-TaTPR1.1-4B, DBD-TaTPR1.1-4D, DBD-TaTPR1.2-7A, DBD-TaTPR1.2-7B, or DBD-TaTPR1.2-7D…” Using “expression of …”, we always mean the expression of genes and should use italics here. Or just delete “the expression of” to refer the protein.
- Response: Thank you very much. “the expression of” has been replaced with “the presence of” in the revised manuscript according to referees’ comments.
- Line 696: Didn’t see any instruction about haustorium index analysis. Also, I suggest giving a brief introduction about the MI index in “4.3. BSMV-mediated gene silencing and microcolony index analysis”.
- Response: Many thanks. The reference for HI and MI analyses have been cited in the revised section of “Materials and Method”.
- Are there any figures of wheat leaves between OE/VIGS and the control? A picture of leaves may give reader a directly impression about how the resistance or susceptibility is.
- Response: Thank you for this suggestion. Single cell transient gene expression/silencing assays and VIGS were performed here, and wheat leaves infected by B.g. tritici were examined by microscope to measure the MI index and HI index in this study.
- TaTPR related to auxin. Overexpression of TaTPR could enhance resistance. Does the overexpression make any yield difference?
- Response: Many thanks. Single cell transient gene expression assays were employed to overexpress TaTPR1 in this study, and the yield difference could not be analyzed at present. However, we will employ the stable transgenic technology to overexpress TaTPR1 and analyzed its potential effect on the wheat yield in the future. This point has been discussed in the revised Discussion Section.
- The susceptible genes in the hosts often play positive or critical roles in plant development, and silencing or deletion of those S genes may have negative effects on plant development or yield. Is there any fitness cost of TaDND?
- Response: Thank you for this suggestion. Single cell transient gene silencing assays and VIGS were employed to silence TaDND1/2 in this study, and the yield penalty could not be analyzed in this situation. W will silence TaDND1/2 by stable transgenic technology and examine its potential effect on the wheat yield in the future. This interesting point has been mentioned in the revised Discussion Section.
Reviewer 3 Report (New Reviewer)
Comments and Suggestions for Authors
After reading the manuscript entitled: Wheat Transcriptional Corepressor TaTPR1 Suppresses Susceptibility Genes TaDND1/2 and Potentiates Post-Penetration Resistance against Blumeria graminis forma specialis tritici, by Pengfei Zhi, Rongxin Gao Wanzhen Chen and Cheng Chang. I consider the manuscript to be relevant to the academic community researching the subject.
Title: Line 4 and 5 - forma specialis, should not be italicized as it is a species variation;
Abstract: Extremely confusing. Authors need to make clear the research question/objective to be answered throughout the manuscript, as well as making clear the methodologies used, the main results and the conclusion of the research;
Keywords: Authors need to pay more attention to the keywords, as they are all present in the title of the manuscript.
Introduction: Line 84 and 95: put the author who described the species followed by the family to which the plant belongs.
Lines 103; 104; 106 and 107: include the name of the author who identified the pathogens; Arabidopsis thaliana (L.) Heynh.
The authors need to make it clear what the aim of the work is at the end of the introduction. The introduction contains confusing information. I suggest rewriting the introduction to make the text clearer and more fluid;
Results: I recommend inserting the figures throughout the text, so that the reader has a better view of your results;
Line 187 and 262: statistical analysis was carried out. How was it carried out? This should be clear in the materials and methods section;
The discussion: Review the discussion, there are many paragraphs that fall into the results section. Write this section as a long text and not in topics;
Methodology: Authors need to pay attention and adapt the methodologies to the results found;
Supplementary material: Authors need to be careful. Figure legends are not a place to describe methodology;
References: Authors need to standardize references and check scientific names that are not in italics, and also the DOI registration number.
Author Response
Reviewer 3 Comments for the Author...
After reading the manuscript entitled: Wheat Transcriptional Corepressor TaTPR1 Suppresses Susceptibility Genes TaDND1/2 and Potentiates Post-Penetration Resistance against Blumeria graminis forma specialis tritici, by Pengfei Zhi, Rongxin Gao Wanzhen Chen and Cheng Chang. I consider the manuscript to be relevant to the academic community researching the subject.
- Response: Thank you very much for these encouraging comments. We have made extensive revision of this manuscript according to referees’ comments. Hopefully, this version could meet the standard for publication.
Title: Line 4 and 5 - forma specialis, should not be italicized as it is a species variation;
- Response: Thank you for this suggestion. According to previous papers (Nat. Genet. 2013, 45(9):1092-6. doi: 10.1038/ng.2704.; Theor Appl Genet. 2021, 134(3):909-921. doi: 10.1007/s00122-020-03741-7.), italicized “Blumeria graminis forma specialis tritici” is widely accepted by the academic community researching wheat powdery mildew disease.
Abstract: Extremely confusing. Authors need to make clear the research question/objective to be answered throughout the manuscript, as well as making clear the methodologies used, the main results and the conclusion of the research;
- Response: Many thanks. The sentence “Topless-related 1 (TPR1) corepressor regulates plant immunity, but its roles in regulating wheat resistance against powdery mildew remains to be disclosed” described the research question/objective to be answered throughout the manuscript. “Transient overexpression of…, while silencing of” described the genetic approaches employed in this study. The conclusion of the research was summarized as the sentence “Collectively, these results implicated that TaTPR1 positively regulates wheat post-penetration resistance against powdery mildew probably via suppressing S genes TaDND1 and TaDND2.”
Keywords: Authors need to pay more attention to the keywords, as they are all present in the title of the manuscript.
- Response: Many thanks. The keywords “wheat; transcriptional corepressor; susceptibility gene; Blumeria graminis forma specialis tritici; TPR1; DND1; DND2” were employed for the convenience of paper searching.
Introduction: Line 84 and 95: put the author who described the species followed by the family to which the plant belongs.
- Response: Thank you for this suggestion. Triticum aestivum L. was employed in this study.
Lines 103; 104; 106 and 107: include the name of the author who identified the pathogens; Arabidopsis thaliana (L.) Heynh.
- Response: Many thanks. Arabidopsis thaliana (L.) Heynh. was employed in the revised manuscript.
The authors need to make it clear what the aim of the work is at the end of the introduction. The introduction contains confusing information. I suggest rewriting the introduction to make the text clearer and more fluid;
- Response: Thank you for this suggestion. Sentences “However, the potential function of wheat TPR1 homologs in the regulation of wheat–B.g. tritici interaction is poorly understood.” and “However, whether and how the wheat DND1 and DND2 homologs regulate the powdery mildew resistance remains unknown.” described the research question/objective to be answered throughout the manuscript.
Results: I recommend inserting the figures throughout the text, so that the reader has a better view of your results;
- Response: Thank you very much. Figures are currently placed at the end of the text for the convenience of review. After acceptance, this manuscript would underdo typesetting and these figures would be placed in the main text of the manuscript then.
Line 187 and 262: statistical analysis was carried out. How was it carried out? This should be clear in the materials and methods section;
- Response: Many thanks. The reference for statistical analysis have been cited in the revised section of “Materials and Method”.
The discussion: Review the discussion, there are many paragraphs that fall into the results section. Write this section as a long text and not in topics;
- Response: Thank you for this suggestion. Discussion arranged in topics is more readable and logic than a long text.
Methodology: Authors need to pay attention and adapt the methodologies to the results found;
- Response: Thank you very much. The section of “Materials and Method” has been revised.
Supplementary material: Authors need to be careful. Figure legends are not a place to describe methodology;
- Response: Thank you for this suggestion. A brief mention of methods employed in the generation of Figure would facilitate the authors’ understanding.
References: Authors need to standardize references and check scientific names that are not in italics, and also the DOI registration number.
- Response: Many thanks. References have been formalized according to the IJMS template.
Round 2
Reviewer 3 Report (New Reviewer)
Comments and Suggestions for Authors
Dear Authors
The recommendation to change the keywords was made to make the article easier to find. The words mentioned by the authors are in the title, so there is no reason why they should also appear in the keywords.
I would ask the authors to present in detail how the data was analyzed, such as the number of repetitions, the acceptable P-value, the statistical test and other information. It doesn't make sense to put it in the figure legend and not detail it in the material and methods section.
Author Response
Reviewer 3 Comments for the Author...
The recommendation to change the keywords was made to make the article easier to find. The words mentioned by the authors are in the title, so there is no reason why they should also appear in the keywords.
- Response: We fully agree with the Reviewer. The keywords have been revised to be distinct from the words in the title.
I would ask the authors to present in detail how the data was analyzed, such as the number of repetitions, the acceptable P-value, the statistical test and other information. It doesn't make sense to put it in the figure legend and not detail it in the material and methods section.
- Response: We thank the Reviewer for this excellent suggestion. We have rephrased the material and methods section to include the detailed information about the data analysis in the revised manuscript. Related description has been removed from the figure legend.
Round 3
Reviewer 3 Report (New Reviewer)
Comments and Suggestions for Authors
The authors have responded to suggestions.
This manuscript is a resubmission of an earlier submission. The following is a list of the peer review reports and author responses from that submission.
Round 1
Reviewer 1 Report
Comments and Suggestions for Authors
Zhi et al conducted a study to explore the functions of TPR1 in the regulation of wheat B. g. tritici interaction. The manuscript provides information. However, there are few crucial points that if considered will increase the accuracy, value of the manuscript and may be readability.
-First of all, please add the benefit or the future prospect of the work, if there is any to the abstract and introduction.
-Please thoroughly explain the ‘plant and pathogen materials’. Please the information on the used wheat genotype. Was it tolerant or susceptible to B.g. tritici? When B.g. tritici were kept on the leaves of Yannong 999 wheat plants? Please explain how the trial was conducted. What was the growth medium for both wheat and Arabidopsis, soil or nutrient solution? The experimental setup is not clear. Please write it thoroughly.
-Please provide information on the quality of the isolated RNAs. Their gel picture or RIN values. You can cite proper studies while explaining the quality of the isolated RNA. For example, https://www.mdpi.com/2073-4395/13/3/631 or https://www.mdpi.com/2073-4395/12/10/2421. RNA quality is important for such studies. Any contamination may affect the reliability of the obtained results.
-Please do add a conclusion to the manuscript.
I do believe that the manuscript can be accepted once the authors address the mentioned points and enrich the manuscript with the crucial information.
Comments on the Quality of English LanguageMinor editing of English language required
Reviewer 2 Report
Comments and Suggestions for Authors
The article “Wheat Transcriptional Corepressor TPR1 Suppresses Susceptibility Genes TaDND1/2 and Potentiates Post-Penetration Resistance against Blumeria graminis forma specialis tritici” (ijms-2679116) studies the role of a transcriptional corepressor on wheat resistance to powdery mildew. The research appears to be well-conducted, but there are some mistakes.
Overall, the manuscript text should have line numbers on the right side of the page to facilitate the review process, but these are missing throughout the manuscript.
Throughout the manuscript there are also some minor English writing mistakes. The manuscript should be carefully checked to correct all these.
Introduction
In the manuscript there are some English writing mistakes. For example in the first paragraph of the Introduction, in “developing wheat variety”, “variety” should be replaced by “varieties”.
Third paragraph of the Introduction: “Arabidopsis transcriptional repressor”, add “thaliana” after “Arabidopsis”.
Results
On the figure legends of Figures 1 and 2 (pages 3 and 4) there is text between the figure and the figure legend. This text should be below the figure legend.
Last paragraph of the Results section (page 10): the first sentence should be in the Materials and Methods section.
Discussion
First paragraph of the Discussion: the statement “this evidence implies that TPR1 positively regulates plant immunity in dicots and monocots” seems like an overgeneralization. This research refers to a specific pathogen and other references included also refer to specific pathogens.
Third paragraph of the Discussion (page 11): the sentence regarding Arabidopsis dnd mutants and its resistance to pathogens requires further explanation regarding its relationship with the current research in the manuscript.
Last paragraph of the Discussion (page 11): the authors should better clarify the reference to HDAC and its association with wheat defenses to powdery mildew and its “unknown” association to wheat TaTPR1 proteins.
Materials and Methods
Primers (pages 12 and 13) could be listed in a supplementary Table.
Comments on the Quality of English LanguageEnglish writing mistakes throughout the manuscript. Please check the text carefully.